# Effects of Dietary Tannic Acid on Obesity and Gut Microbiota in C57BL/6J Mice Fed with High-Fat Diet

**DOI:** 10.3390/foods11213325

**Published:** 2022-10-23

**Authors:** Jiangmin Fang, Lirong Zeng, Yalun He, Xiong Liu, Tongcun Zhang, Qiong Wang

**Affiliations:** College of Life Sciences and Health, Wuhan University of Science and Technology, Wuhan 430065, China

**Keywords:** obesity, dietary tannic acid, oxidative stress, glycolipid metabolism, gut barrier, gut microbiota

## Abstract

Dietary tannic acid, as a natural polyphenolic, has many important biological activities. This study aimed to investigate the effect of dietary tannic acid on obesity and gut microbiota in mice with a high-fat diet. Male C57BL/6J mice fed a high-fat diet were treated with dietary tannic acid for eight weeks. Results showed that dietary tannic acid reduced the body weight gain, regulated glycolipid metabolism, improved the insulin resistance, and attenuated the liver oxidative stress in high-fat diet-fed mice. Moreover, both dietary tannic acid intervention groups repaired the gut barrier damage caused by a high-fat diet, especially in the 50 mg/kg/d dietary tannic acid intervention group. Interestingly, the effect of dietary tannic acid on serum endotoxin lipopolysaccharide (LPS) content was correlated with the abundance of the LPS-producing microbiota. In addition, dietary tannic acid altered the abundance of obesity-related gut microbiota (Firmicutes, Bacteroidetes, *Bacteroides*, *Alistipes*, and *Odoribacter*) in the 150 mg/kg/d dietary tannic acid intervention group, while it was not effective in the 50 mg/kg/d dietary tannic acid intervention group. These findings suggested the potential effect of dietary tannic acid for the prevention and control of obesity.

## 1. Introduction

Obesity has become a pressing public health problem which can increase the risk of many chronic diseases, such as type 2 diabetes and fatty liver and cardiovascular diseases [1,2]. It is characterized by the disorder of glucolipid metabolism, insulin resistance, oxidative stress, and inflammatory response [3]. Therefore, preventing obesity is a health challenge in modern society.

Reports showed that dietary polyphenols contained in the daily diet have the ability to prevent and control the occurrence of obesity [4,5,6,7]. Dietary tannic acid is a natural polyphenolic compound commonly found in plants, which has always been listed as an anti-nutritional factor [8]. Further research found that dietary tannic acid has many important physiological activities, such as antioxidant, anti-inflammatory, antibacterial, lowering blood glucose and blood lipid activities, etc. [9,10,11]. Due to a large number of hydroxyl groups in its structure, dietary tannic acid can interact with starch, polysaccharides, and proteins in the diet, as well as the digestive enzymes, to form insoluble complexes, which reduce the digestibility of food in the diet [12,13]. In addition, tannic acid can affect the glycolipid metabolism signaling pathways in 3T3-L1 cells [14,15]. It has a positive effect on the translocation of GLUT4 to the cell membrane, and promotes the phosphorylation of insulin receptor IR and Akt, thus promoting glucose transport [14]. Meanwhile, tannic acid can inhibit adipocyte differentiation by inhibiting the expression of FAS and PPARγ [15]. However, the effect of dietary tannic acid on obesity remains unclear.

Growing evidence suggests that obesity is accompanied by alterations in the gut microbiota [16,17]. The gut microbiota dysbiosis induced by obesity will damage the intestinal integrity, which leads to endotoxin lipopolysaccharide (LPS) being released into the blood, thus resulting in metabolic inflammation in obese mice [18,19,20]. Research showed that dietary polyphenols have potential health effects on gut microbiota [21]. Additionally, dietary tannic acid is reported to promote the proliferation of probiotics and inhibit the growth of various harmful intestinal bacteria [22,23]. More research showed that dietary tannic acid can improve the gut barrier integrity and gut microbiota in weaned piglets [24,25]. However, the role of gut microbiota in the anti-obesity effect of dietary tannic acid has not been studied. Therefore, this study aims to explore the impact of dietary tannic acid on obesity with respect to improving glycolipid metabolism, inflammation, and gut microbiota, which provide new ideas for the prevention and control of obesity.

## 2. Materials and Methods

### 2.1. Reagents and Materials

Dietary tannic acid was purchased from Sigma-Aldrich (St. Louis, MO, USA). Assay kits of total cholesterol (TC), triacylglycerol (TG), high-density-lipoprotein cholesterol (HDL-C), low-density-lipoprotein cholesterol (LDL-C), liver malondialdehyde (MDA), total superoxide dismutase (SOD), alkaline phosphatase (AKP), alanine transaminase (ALT), and aspartate transaminase (AST) concentrations were purchased from Nanjing Jiancheng Bioengineering Institute (Nanjing, China). ELISA kits of insulin, leptin, lipopolysaccharides (LPS), serum inflammatory cytokines (TNF-α and IL-6), and immunoglobulins (IgA, IgG, IgE, and IgM) were purchased from Wuhan Ilerite Biotechnology Co., Ltd., (Wuhan, China). Antibodies, including β-actin, AMPKα, *p*-AMPK, PPARγ, ZO-1, and Occludin, were purchased from Beyotime Biotechnology (Shanghai, China).

### 2.2. Animals and Diets

Animal experiments were performed in accordance with the guidelines of the medical ethics committee of the School of Life Science and Health, Wuhan University of Science and Technology. Male C57BL/6J mice (7 weeks, about 19–22 g) were purchased from Shanghai Model Organisms. The caloric percentages of fat, protein and carbohydrate in the 60% high-fat diet (code TP23300) were 60%, 19.4% and 20.6%, respectively. The percentage of fat and protein in AIN93G standard feed (code: LAD3001G) were 16.7% and 19.3%, respectively. All the feeds were purchased from Trophic Animal Feed Technology Co., Ltd., Jiangsu, China. Mice were randomly divided into four groups (*n* = 9): normal diet group (ND group), high-fat diet group (HFD group), 50 mg/kg/d dietary tannic acid intervention group (HFD_TA 50 group) and 150 mg/kg/d dietary tannic acid intervention group (HFD_TA150 group). Mice in the ND and HFD groups were intragastrically given normal saline at the dose of 10 mL/kg/d, while mice in the intervention groups were intragastrically assigned dietary tannic acid solution at the amount of 50 mg/kg/d and 150 mg/kg/d at 10 am every day. During the experiment intervention period, animals were kept under an indoor temperature of 20–25 ℃ for 12 h light every day and allowed to eat and drink freely.

### 2.3. Sample Collection

After intervention for eight weeks, mice in each group were fasted for 12 h without the prohibition of water. The fasting blood glucose was detected with a Roche glycemic meter. The mice were intraperitoneally injected with glucose solution at a dose of 2 g/kg, and blood was collected from the tail tip at 30, 60, 90, and 120 min for the glucose tolerance test. The whole blood was collected from the orbital plexus to obtain serum. Then, the mice were sacrificed to obtain the liver, abdominal fat, subcutaneous fat, pancreas, colonic tissues, and colon contents. The abdominal fat tissues of the mice were weighed and compared with their respective body weight to obtain the corresponding body fat percentage. A small part of the liver, epididymal adipose, pancreas and colon tissues were fixed in 4% formaldehyde solution for histopathological examination. The remaining tissues were quickly frozen in liquid nitrogen and stored at −80 ℃ for further analysis.

### 2.4. Serum and Liver Index Biochemical Analysis

The liver tissue was accurately weighed, and thoroughly ground into liver homogenate in proportion to liver weight (g): saline volume (mL) = 1:9. Then, the supernatant was obtained after being centrifuged at 12,000 r/min for 5 min, and the protein concentration was determined by the BCA method. The concentrations of serum and liver index (TC, TG, HDL-C, LDL-C, MDA, SOD, ALT, AST, and AKP) were determined using commercial kits (Nanjing Jiancheng Bioengineering Institute, Nanjing, China). The contents of serum insulin, leptin, lipopolysaccharides (LPS), inflammatory cytokines (TNF-α and IL-6), immunoglobulins (IgA, IgG, IgE, and IgM) were determined by ELISA kits (Wuhan Ilerite Biotechnology Co., Ltd., Wuhan, China).

### 2.5. Histological Examination

Fresh liver, epididymal adipose, pancreas and colon tissues were fixed in 4% formaldehyde for 24 h. The paraffin-embedded tissue was sectioned according to the slice thickness requirement of 4 μm on the slicer. The sections were stained with H&E and observed using an optical microscope (Nikon Eclipse TE2000-U, Nikon, Japan).

The OCT-embedded liver tissue was placed in a frozen section machine, and sectioned according to the slice thickness requirement of 10 μm on the slicer. The sections were stained with Oil Red O for 10 min, and slightly differentiated with 75% alcohol. Then, the sections were counter-stained with haematoxylin for 4 min. The adipose area was measured according to the scanning results of Oil Red O staining slices by using ImageJ software.

### 2.6. RT-qPCR Analysis

Total RNA was isolated from the liver or colon tissues and then reverse transcribed to cDNA. Real-time qPCR was performed with *PPAR-**γ*, *SREBP*, *LPL*, *FAS*, *ACC*, *AMPK*, *CPT-1*, *IL-6*, *TNF-**α*, *ZO-1*, and *Occludin* as target genes. The primers used in this study are listed in Appendix A, which were synthesized by Jin Weizhi Biological Technology Co., Ltd., Suzhou, China.

### 2.7. Western Blot Analysis

Proteins were extracted from the liver or colon tissue, separated using SDS-PAGE, then transferred to PVDF membranes. The membranes were incubated overnight at 4 °C using rabbit polyclonal antibodies against β-actin (Catalog no. AF0003), AMPKα (Catalog no. AF1627), *p*-AMPK (Catalog no. AF2677), PPARγ (Catalog no. AF7797), ZO-1 (Catalog no. AF8394), and Occludin (Catalog no. AF7644) (Beyotime Biotechnology, Shanghai, China). The corresponding secondary antibody and ECL reagent were used to detect the target protein immobilized on the membrane.

### 2.8. 16S rRNA Sequencing

Total genomic DNA was extracted with QIAamp DNA Stool Mini Kit from colon contents. PCR amplification was performed with variable region V4-V5 of 16S rRNA gene with a special primer (515F 5′-GTGCCAGCMGCCGCGGTAA-3′ and 926R 5′-CCGTCAATTCMTTTGAGTTT-3′). PCR products were recovered by AxyPrepDNA Gel Recovery Kit and sequenced by Illumina high-throughput sequencing at Shanghai Tinygene Biotechnology Co., Ltd. (Shanghai, China). Operational taxonomic units (OTUs) clustering analysis was generated using Usearch. Alpha diversity analysis were calculated using QIIME and Mothur. Beta diversity analysis were calculated using the vegan package in R(3.4.1).

### 2.9. Statistical Analysis

Data were statistically analyzed and plotted by GraphPad Prism 6. Results were expressed as mean ± SEM. Statistical significance was measured using one-way analysis of variance (ANOVA) with Duncan’s range tests. Different letters between any two groups indicated that there was significant difference between the two groups (*p* < 0.05).

## 3. Results

### 3.1. Dietary Tannic Acid Prevents Obesity in Mice with a High-Fat Diet

The intervention of dietary tannic acid significantly reduced the body weight with a decrease of weekly food intake (*p* < 0.05, Figure 1B–D). Besides, the weight of liver tissue was significantly reduced only in the HFD_TA150 group (*p* < 0.05, Figure 1E). Moreover, dietary tannic acid intervention decreased the subcutaneous fat, abdominal fat, and body fat percentage. In particular, the abdominal fat and body fat percentage in the HFD_TA150 group were significantly different from the HFD group (*p* < 0.05, Figure 1E,F). To investigate the effect of dietary tannic acid on the accumulation of fat in mice fed with a high-fat diet, the histopathological morphologies of epididymal adipose tissue were examined by using H&E staining (Figure 1H). As shown in Figure 1G, the area of adipose tissue cells of mice in the HFD group was significantly increased compared to the ND group, while it was decreased by 42.72% and 62.63% in dietary tannic acid intervention groups.

### 3.2. Dietary Tannic Acid Improves the Glucolipid Metabolism in Mice with a High-Fat Diet

Compared with the HFD group, the dietary tannic acid intervention significantly reduced the levels of fasting blood glucose and improved the ability of glucose tolerance (Figure 2A–C). Consistent with this result, the levels of serum insulin and leptin were significantly decreased in the dietary tannic acid intervention group (Figure 2D). To further explore the influence of dietary tannic acid on insulin secretion, the structure of pancreatic tissue is shown in Figure 2J. Compared with the ND group, HFD feeding decreased the number of pancreas islets, while dietary tannic acid could improve this phenomenon. Moreover, the serum lipid levels (TC, TG, HDL-C, and LDL-C) in the dietary tannic acid intervention groups were significantly reduced in mice fed with a high-fat diet (*p* < 0.05, Figure 2E), indicating that the intervention of dietary tannic acid could improve the dyslipidemia of mice induced by a high-fat diet.

To explore the mechanism of dietary tannic acid on blood glucose and lipids, the expression of related genes involved in glycolipid metabolism was detected. Results showed that dietary tannic acid significantly increased the mRNA expression of *AMPK*, while decreasing the mRNA expression of *PPAR**γ*, *SREBP*, *ACC*, *FAS*, and *LPL* in the hepatic tissue of HFD-fed mice (Figure 2F). Correspondingly, the ratio of *p*-AMPK/AMPK and the protein expression of phosphorylated AMPK were significantly increased, while the protein expression of AMPK and PPARγ were decreased in the dietary tannic acid intervention groups when compared to the HFD group (Figure 2G–I).

### 3.3. Dietary Tannic Acid Attenuates Liver Lipid Accumulation, Oxidative Stress, and Liver Toxicity in Mice with a High-Fat Diet

There is a strong correlation between body obesity and the occurrence of fatty liver. To further explore the effect of dietary tannic acid on liver lipid accumulation, the Oil Red staining of liver tissue is shown in Figure 3A. Compared with the ND group, the content of fat droplets in liver cells was significantly increased in the HFD group. After intervention with dietary tannic acid, the content of fat droplets and the fat accumulation in liver were significantly reduced. Besides, compared with the ND group, the TC and TG content in the liver tissue of mice with obesity caused by a high-fat diet were significantly increased (*p* < 0.05), while dietary tannic acid intervention reversed this phenomenon (Figure 3B).

Reports showed that obesity resulted in the oxidative stress in the liver, which led to the accumulation of reactive oxygen species, and finally damaged the liver cells [26]. Compared with the HFD group, the content of MDA in the liver tissue was reduced by 54.01% and 60.73% in the HFD_TA50 and HFD_TA150 group, while the content of SOD in the liver tissue was increased by 5% and 8.57% (Figure 3C). This indicated that dietary tannic acid intervention alleviated the oxidative stress in the liver. Correspondingly, compared with the HFD group, both dietary tannic acid interventions significantly decreased the ALT activity in liver tissue by 47.51% and 48.56% in the HFD_TA50 and HFD_TA150 groups, respectively. However, only dietary tannic acid intervention in the HFD_TA150 group significantly reduced the AST and AKP activity in liver tissue (Figure 3D). Meanwhile, the content of ALT and AST in serum were both significantly reduced after the dietary tannic acid intervention (Figure 3E). These results indicated that dietary tannic acid was effective in preventing liver injury caused by a high-fat diet.

### 3.4. Dietary Tannic Acid Reduces Systemic Inflammation in Mice with a High-Fat Diet

The changes in serum IgE, IgM, IgG, and IgA contents were detected by the ELISA kit (Figure 3F). Compared with the HFD group, dietary tannic acid intervention showed a reduction in the content of the four serum immunoglobulins (IgE, IgM, IgG, and IgA). Additionally, the level of serum LPS was significantly increased in the HFD group than in the ND group (*p* < 0.05), which it was reduced by dietary tannic acid intervention (Figure 3G). The content of proinflammatory cytokines (TNF-α and IL-6) in the serum, liver and colon tissue were markedly reduced by dietary tannic acid intervention (Figure 3H). Moreover, the liver tissue of mice with a high-fat diet showed a series of vacuoles with different sizes, cell integrity destruction and inflammatory infiltration. As expected, the liver damage was improved by the dietary tannic acid intervention (Figure 3I). Particularly, the liver tissue in the HFD_TA150 group presented a clear, complete, and less inflammatory infiltration structure.

### 3.5. Dietary Tannic Acid Ameliorates the Gut Barrier Damage Caused by High-Fat Diet

H&E staining revealed that high-fat diet caused severe damage to the colonic structure in the HFD group, and it was partially recovered after the dietary tannic acid intervention (Figure 4A). Compared with the ND group, there were significant reductions in crypt depth and goblet cells in the HFD group, while dietary tannic acid intervention significantly attenuated these reductions (Figure 4B,C). Consistent with this result, the expression of TJ proteins (ZO-1 and Occludin) in the colonic tissue were significantly reduced in the HFD group, and it recovered with the addition of dietary tannic acid (Figure 4D). In particular, dietary tannic acid intervention in the HFD_TA50 group showed better improvement in the expression of TJ proteins in colon tissues. Next, we detected the concentration and localization of TJ proteins in colon tissues by using immunofluorescence. As shown in Figure 4E, the fluorescence intensity of ZO-1 and Occludin in the HFD group were almost completely destroyed in the mucous layer, and only existed in the muscular layer and serous layer. However, dietary tannic acid intervention recovered the fluorescence intensity and localization of ZO-1 and Occludin, which showed a protective effect on the gut barrier.

### 3.6. Dietary Tannic Acid Improves the Structure of Gut Microbiota in Mice with High-Fat Diet

Compared to the HFD group, dietary tannic acid intervention significantly increased the ace, chao, PD_whole_tree, Shannon diversity, and Sobs index, while decreasing the Simpson diversity (*p* < 0.05, Figure 5A). This suggested that dietary tannic acid could improve the gut microbiota structure and optimize the species diversity index. Additionally, PCA, PCoA and NMDS reflected the differences and distances among groups. As shown in Figure 5B, the distance between the dietary tannic acid intervention groups and the HFD group was clearly separated, while the distance between dietary tannic acid intervention groups and ND group was closer. Venn diagram analysis showed the similarity and overlap of OTU number composition, while heatmaps at the phylum level reflected the similarity and difference of community composition in the four groups (Figure 5C,D).

At the phylum level, the relative abundance of Firmicutes and Proteobacteria were significantly increased, while the abundance of Bacteroidetes was significantly decreased in the HFD group. Dietary tannic acid significantly decreased the relative abundance of Proteobacteria and Firmicutes, while increasing the abundance of Bacteroidetes in the HFD_TA150 group (*p* < 0.05). However, the abundance of Firmicutes, Bacteroidetes and Proteobacteria showed no significant differences in the HFD_TA50 group when compared to the HFD group. The ratio of Firmicutes/Bacteroidetes was significantly higher in the HFD group than that in the ND group (*p* < 0.05). The ratio of Firmicutes/Bacteroidetes were both decreased in dietary tannic acid intervention groups, but only showed significantly decrease in the HFD_TA150 group (*p* < 0.05, Figure 6A,B). At the genus level, dietary tannic acid significantly decreased the abundance of *Desulfovibrio*, while increased the abundance of *Bacteroides*, *Alistipes*, and *Odoribacter* in the HFD-TA150 group. Similar to the phylum level, the HFD_TA50 group showed no significant difference with the HFD group on the above four species at the genus level, except for *Odoribacter* (Figure 6C,D).

### 3.7. Correlation between the Gut Microbiota and Obesity

The correlations between gut microbiota (Firmicutes, Bacteroidetes, Proteobacteria, Cyanobacteria, *Bacteroides*, *Alistipes*, *Desulfovibrio*, and *Odoribacter*) and obesity-related indexes (body weight, abdominal fat, and subcutaneous fat) were revealed through Spearman correlation analysis (Figure 7). Among these taxa, Firmicutes, Proteobacteria, and *Desulfovibrio* displayed a positive relationship with body weight, abdominal fat and subcutaneous fat, while a negative correlation was shown by Bacteroidetes, Cyanobacteria, *Bacteroides*, *Alistipes*, and *Odoribacter*.

## 4. Discussion

The addition of dietary tannic acid in the HFD_TA50 and HFD_TA150 groups showed a similar reduction in body weight and weekly food intake (Figure 1B–D). Reducing food intake may be one of the more important reasons for the prevention of dietary tannic acid in obesity. Both dietary tannic acid interventions significantly reduced the levels of blood glucose and lipid. In particular, although mice in dietary tannic acid intervention groups had no significant difference in weekly food intake, mice in the HFD_TA50 group showed lower blood glucose and higher blood lipid levels than those in the HFD_TA150 group. This indicated that the two dietary tannic acid intervention groups might have different performances of glycolipid metabolism. Previous studies showed that the AMPK and PPARγ signaling pathways were very important in regulating glycolipid metabolism [27,28,29]. In this study, dietary tannic acid intervention up-regulated the levels of *p*-AMPK, which inhibits the expression of SREBP, ACC, and FAS in the fatty acid synthesis pathway [30,31]. This was confirmed by the mRNA expression levels of *SREBP*, *ACC* and *FAS*. Additionally, dietary tannic acid intervention increased the mRNA expression levels of *CPT-1* in liver tissues of mice fed with a high-fat diet, which could promote the β-oxidation process of fatty acids and inhibit the synthesis of TG in vivo [32]. Moreover, the protein expression of PPARγ, which was the main regulator of adipocyte gene expression and insulin cell signal transduction [33], was decreased by the addition of dietary tannic acid. Therefore, dietary tannic acid may also regulate the metabolic disorders caused by a high-fat diet through the AMPK and PPARγ signaling pathways, thereby inhibiting the formation of obesity.

Mice in the HFD group were accompanied by leptin and insulin resistance, which improved with dietary tannic acid intervention. Reports showed that leptin and insulin resistance play a vital role in the development of obesity [34,35,36]. Additionally, there was a strong positive correlation among insulin resistance, oxidative stress and inflammation [37,38]. Dietary tannic acid, as an antioxidant, effectively improved the liver oxidative damage caused by a high-fat diet (Figure 3A). Furthermore, dietary tannic acid suppressed the inflammatory cytokines in serum, liver and colon (Figure 3H), which correspond with the results that less inflammatory infiltration was observed in the liver and colon tissues (Figure 3I and Figure 4A). As molecular mediators, the inflammatory cytokines play a key role in the immune responses [39]. The immunoglobulin antibodies may attack islet beta cells and insulin-sensitive tissues, putting the body into a state of insulin resistance [40]. In this study, the decreased content of IgA, IgE, IgM and IgG may be part of the reason for the improvement of the islet cells in the pancreas of mice with dietary tannic acid intervention, which finally improved the insulin resistance. These results indicated that dietary tannic acid might be a potential candidate for the treatment of obesity through the improvement of insulin resistance.

Results showed that a chronic high-fat diet significantly reduced the expression of TJ proteins in colon tissue, which destroyed the gut barrier and resulted in the permeation of LPS concentrations in the blood. Both dietary tannic acid intervention groups repaired the gut barrier damage caused by a high-fat diet, especially in the HFD_ TA50 group. It can be concluded that the gut barrier repaired by the dietary tannic acid intervention could block the transformation of LPS from the intestinal tract to the bloodstream. Evidence showed that LPS was the main component of the outer membrane of Gram-negative bacteria cell wall, which could lead to metabolic inflammation [16,20]. Consistent with this result, the relative abundance of the LPS-producing microbiota such as Proteobacteria and *Desulfovibrio* were reduced in the HFD_TA150 group, which may help to regulate inflammation [41,42,43]. This was supported by the reduced inflammatory cytokines in serum, liver and colon. However, it is worth noting that although dietary tannic acid intervention in the HFD_TA50 group showed better improvement in the gut barrier damage, the content of serum LPS in the HFD_TA50 group was similar to the HFD group. Besides, the content of Proteobacteria and *Desulfovibrio* did not significantly reduce in the HFD_TA50 group when compared to the HFD group. These findings implied that the regulation of dietary tannic acid on serum LPS content was correlated with the abundance of LPS-producing microbiota.

Growing evidence has shown that gut microbiota with a high degree of microbial diversity are generally associated with positive health outcomes [44]. The increased gut microbial diversity in both dietary tannic acid intervention groups may contribute to the regulation of obesity. Previous reports have shown that gut microbiota have an important role in adjusting obesity, especially the Firmicutes and Bacteroidetes [45]. The increase in the ratio of Firmicutes/Bacteroidetes has been linked to an increased absorption of calories from foods by gut microbes [46]. In this study, dietary tannic acid treatment in the HFD_TA150 group reversed the abundance of the Firmicutes and Bacteroidetes caused by a high-fat diet, which restored the Firmicutes/Bacteroidetes ratio to that in the ND group (Figure 6B). Besides, the abundances of *Bacteroides*, *Alistipes* and *Odoribacter*, which were reported to negatively correlate with obesity [47,48,49], were upregulated in the HFD_TA150 group. However, dietary tannic acid in the HFD_TA50 group had no significant influence on the abundance of the obesity-related gut microbes. Interestingly, both dietary tannic acid intervention groups showed a similar anti-obesity effect. This result suggested that gut microbial diversity may be more responsible for the regulation of dietary tannic acid on obesity than gut microbial composition in the present study. Moreover, dietary tannic acid in the HFD_TA150 group had a better impact on reducing the weight of adipose tissue and liver lipid accumulation than the HFD_TA50 group. It can be speculated that the regulation of lipid metabolism by dietary tannic acid may be related to the abundance of the obesity-related gut microbes.

## 5. Conclusions

In conclusion, dietary tannic acid in different doses may exert its anti-obesity effect through different pathways, which was closely related to the effect of dietary tannic acid on glycolipid metabolism, inflammation and gut microbiota. However, more research is needed to further explore its precise mechanism of action.

## Figures and Tables

**Figure 1 foods-11-03325-f001:**
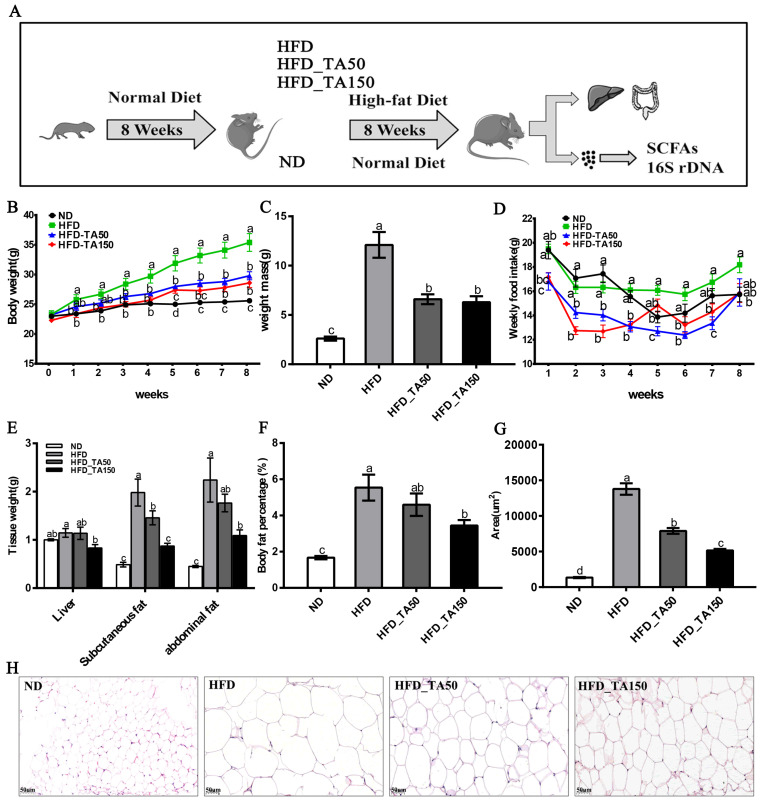
Effect of dietary tannic acid on body weight, fat mass and adipocyte morphology in mice induced by high-fat diet. (**A**) Timeline depicting the treatment of tannic acid, (**B**) body weight, (**C**) weight mass, (**D**) weekly food intake, (**E**) tissue weight, (**F**) body fat percentage, (**G**) lipid drops area, (**H**) epididymal adipose tissue morphology. ND: normal diet, HFD: high-fat diet, HFD_TA50: 50 mg/kg/d tannic acid intervention, HFD_TA150: 150 mg/kg/d tannic acid intervention. Data are expressed as the mean ± SEM. Different letters between any two groups indicated that there was significant difference between the two groups (*p* < 0.05).

**Figure 2 foods-11-03325-f002:**
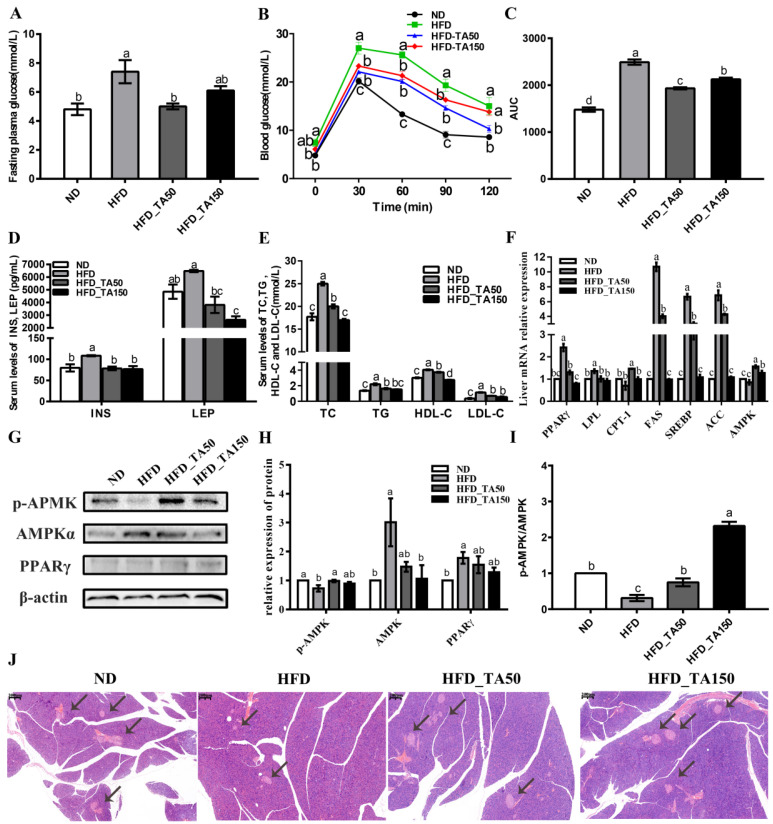
Effect of dietary tannic acid on the glucolipid metabolism in mice induced by a high-fat diet. (**A**) Fasting plasma glucose, (**B**) blood glucose, (**C**) area under curve of OGTT, (**D**) serum INS and LEP, (**E**) serum levels of TC, TG, HDL-c and LDL-c, (**F**) relative expression of mRNA levels in the liver, (**G**) Western blot of hepatic AMPKα, *p*-AMPK and PPARγ, (**H**) relative expression of hepatic AMPKα, *p*-AMPK and PPARγ, (**I**) *p*-AMKP/AMPK, (**J**) H&E staining of pancreatic tissues. Black arrows indicate the pancreas islets. INS: insulin, LEP: leptin, TC: serum cholesterol, TG: serum triglyceride, HDL-c: high-density-lipoprotein cholesterol, LDL-c: low-density-lipoprotein cholesterol. Data are expressed as the mean ± SEM. Different letters between any two groups indicated that there was significant difference between the two groups (*p* < 0.05).

**Figure 3 foods-11-03325-f003:**
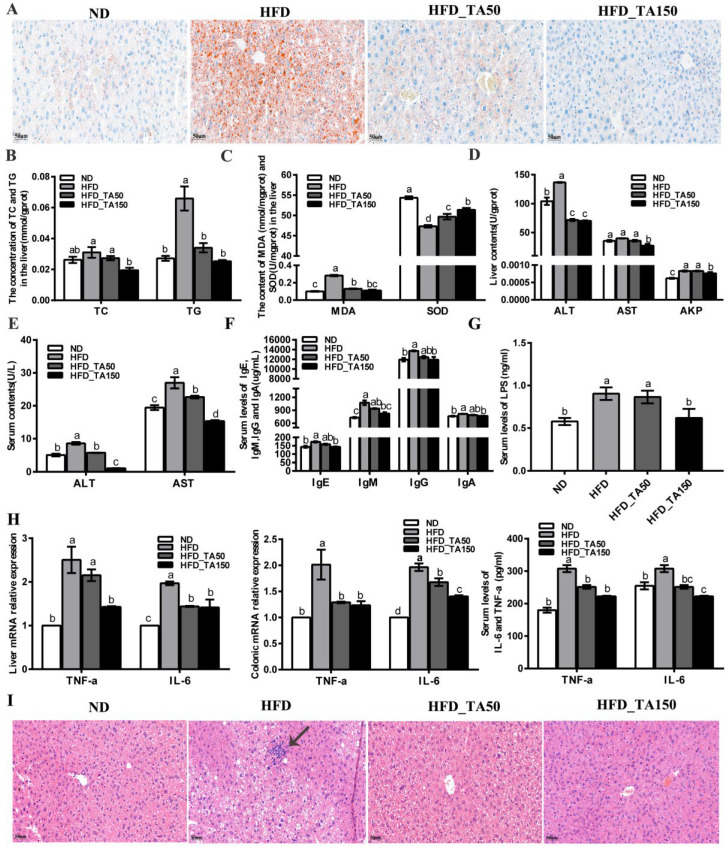
Effect of dietary tannic acid on liver lipid accumulation, oxidative stress, liver toxicity and systemic inflammation in mice induced by high-fat diet. (**A**) Hepatic histopathologic appearance by Oil Red staining, (**B**) the concentration of TC and TG in the liver, (**C**) the content of MDA and SOD in the liver, (**D**) the content of ALT, AST and AKP in the liver, (**E**) the content of ALT and AST in serum, (**F**) the content of IgE, IgM, IgG and IgA in serum, (**G**) the content of LPS in serum, (**H**) the content of TNF-α and IL-6 in serum, liver, and colon, (**I**) hepatic histopathologic appearance by H&E staining. Black arrows indicate the inflammatory infiltration structure. Data are expressed as mean ± SEM. Different letters between any two groups indicated that there was significant difference between the two groups (*p* < 0.05).

**Figure 4 foods-11-03325-f004:**
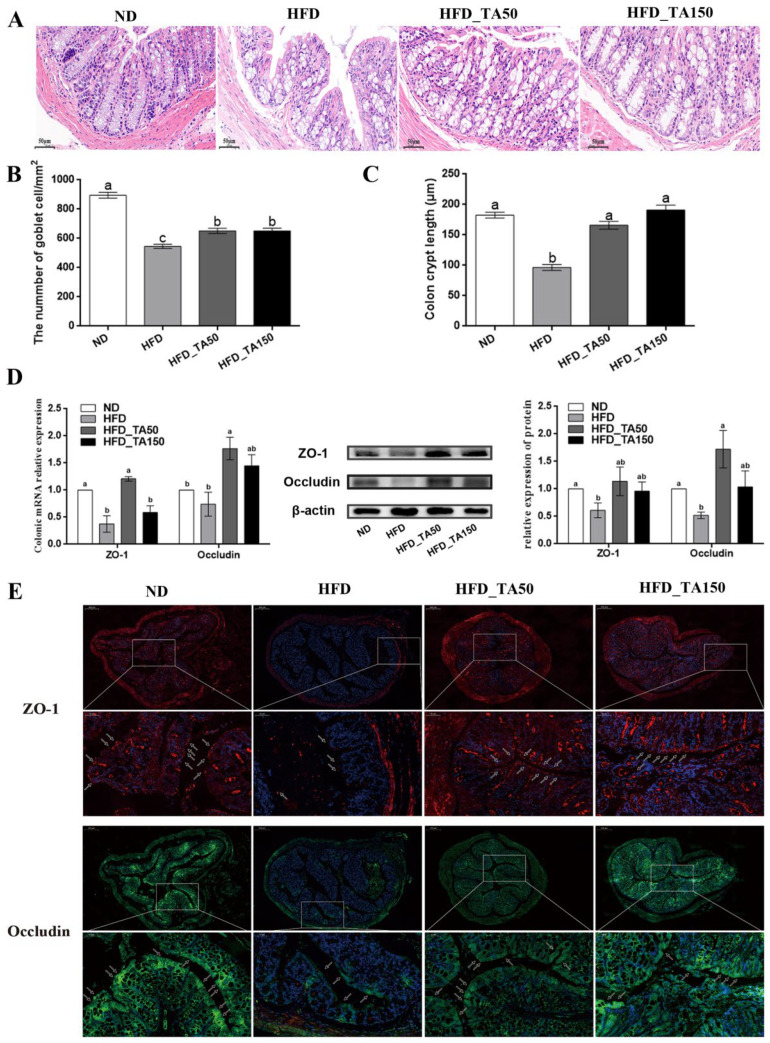
Effect of dietary tannic acid on the intestinal barrier in mice induced by high-fat diet. (**A**) Representative colon H&E staining images, (**B**) the number of goblet cells, (**C**) colon crypt length, (**D**) colon ZO-1 and Occludin mRNA and protein expression measured by qRT-PCR and Western blot, (**E**) representative images of immunofluorescence in colon tissues with antibodies against ZO-1 and Occludin. Data are expressed as mean ± SEM. Different letters between any two groups indicated that there was significant difference between the two groups (*p* < 0.05).

**Figure 5 foods-11-03325-f005:**
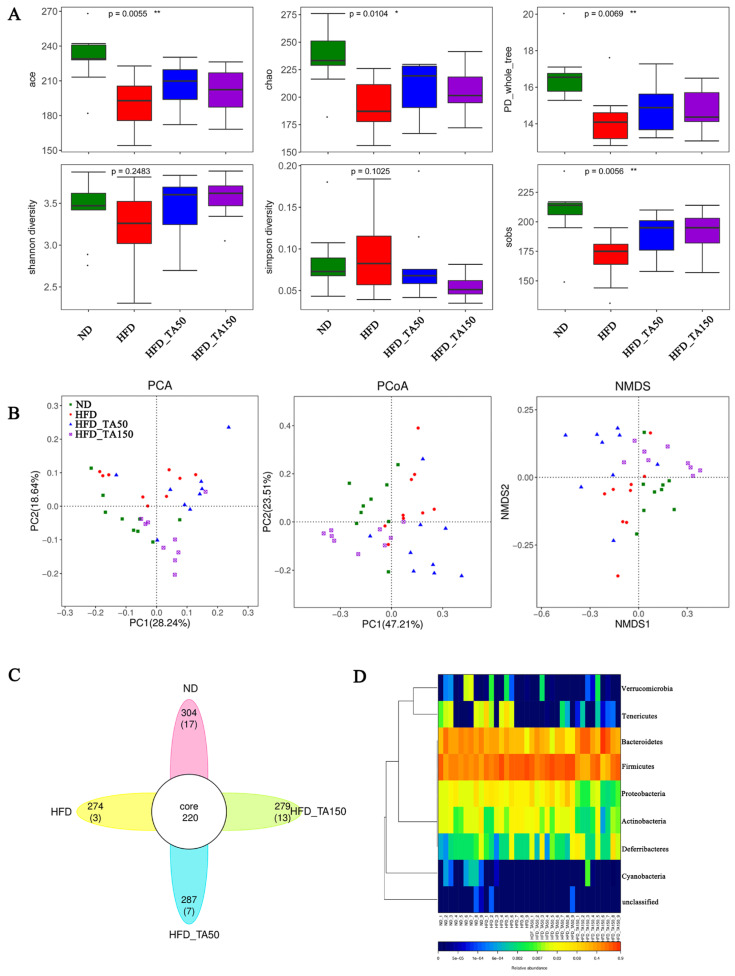
Effect of dietary tannic acid on gut microbial diversity. (**A**) The alpha diversity analysis (ace, chao, PD_whole_tree, Shannon diversity, Simpson diversity, sobs), (**B**) the beta diversity analysis (PCoA, PCA and NMDS), (**C**) Venn diagrams, (**D**) heatmap on phylum level. Data are expressed as mean ± SEM. Significant correlations are marked by * *p* < 0.05 and ** *p* < 0.01.

**Figure 6 foods-11-03325-f006:**
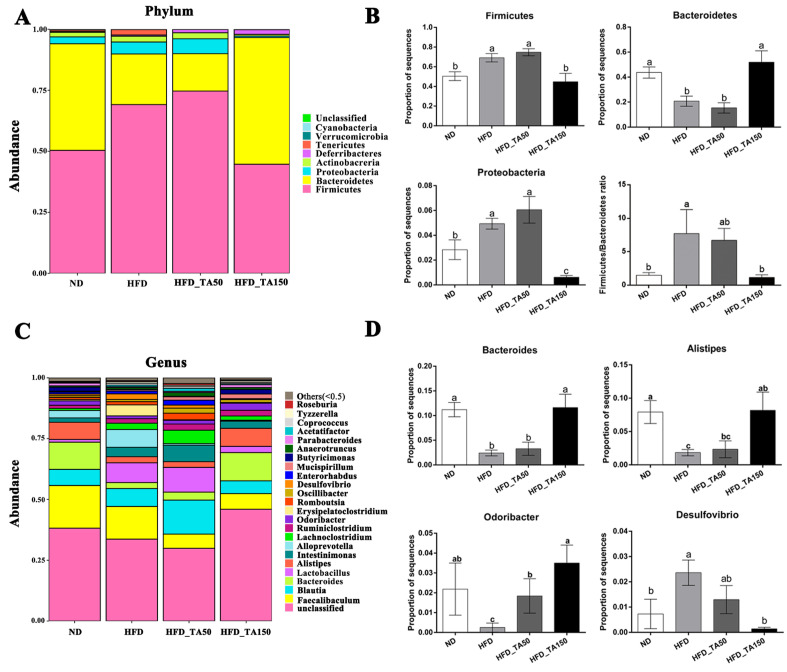
Effect of dietary tannic acid on the composition of gut microbiota at the phylum and genus level. (**A**) Taxonomic distribution at the phylum level, (**B**) relative abundance of Firmicutes, Bacteroidetes, Proteobacteria, and Firmicutes/Bacteroidetes ratio, (**C**) taxonomic distribution at the genus level, (**D**) relative abundance of *Bacteroides*, *Alistipes*, *Odoribacter* and *Desulfovibrio*. Data are expressed as mean ± SEM. Different letters between any two groups indicated that there was significant difference between the two groups (*p* < 0.05).

**Figure 7 foods-11-03325-f007:**
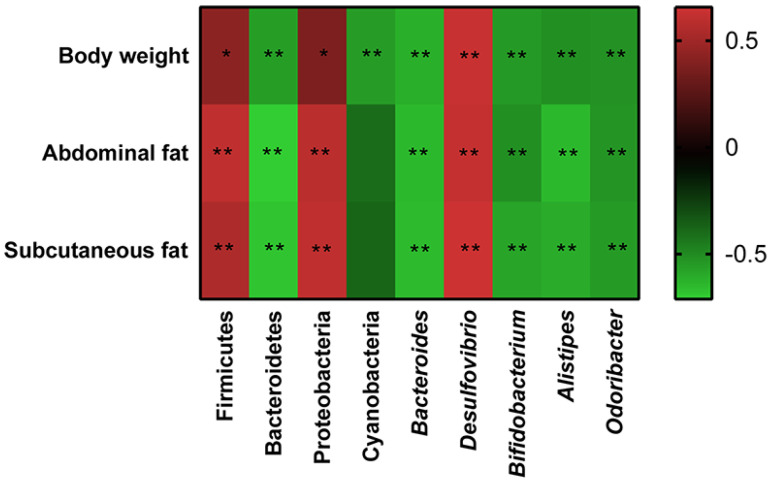
Heatmap of Spearman’s correlation between the gut microbiota and obesity-related indexes. The colors range from blue (negative correlation) to red (positive correlation). Significant correlations are marked by * *p* < 0.05 and ** *p* < 0.01.

## Data Availability

The data are available from the corresponding author.

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
