# Peer review of "Effects of Dietary Tannic Acid on Obesity and Gut Microbiota in C57BL/6J Mice Fed with High-Fat Diet"

_foods, 2022, doi:10.3390/foods11213325_

Round 1
Reviewer 1 Report
The manuscript is well designed and the work is scientifically sound.
Authors should consider either consolidating bar graphs or either way improve the visibility to make the statistics more clear on the figures.
Some reference to interaction of tannic acid with usually food/gut components could be included in addition to current discussion, in light of recent references.
Overall the work is timely and appealing
Author Response
Answer: Thank you very much for your positive comments on our manuscript. We have accepted these suggestions. The font size in Figures have been modifed to make the statistics more clear. Besides, the color of the signs for differentiating the groups in Figure 1B, 1D, and 2B were modified to be more distinctive. Moreover, two references about the effect of tannic acid on gut health were added in the introduction section (Page 2, Line 53-54).

Reviewer 2 Report
In the present study, authors investigated effect of tannic acid on HFD induced disorder in mice.
I think this study have adequately conducted and well written, although detail of mechanism is unclear.
Information shown in this manuscript can help to understand effect of tannic acid on high fat diet induced disorder.
I have several minor comments.
1. In abstract. Please inform HFD_TA50 and HFD_TA150 more detail. It is difficult to understand.
2. In abstract. There is only correlation in serum LPS and abundance of LPS-producing microbiota. It is an overestimate to conclude “serum LPS content was more influenced”.
3. Introduction line 39. How researcher investigate effect of dietary TA in 3T3-L1 cells. I think authors can delete “dietary”.
4. Method. Please provide detail information of antibody (ex. Catalog no., epitope no., or something).
5. Method. Please provide detail information of HFD.
6. Method. Please mention why authors used these concentrations of TA (50 mg/kg/d and 150 mg/kg/d)
7. Method line 88. This method is not “oral” glucose tolerance test.
8. Method line 90. What kind of abdominal fat was used? Epididymal, perirenal, or others?
9. Method. Please provide detail information to measure lipid concentration in liver, especially in extraction of lipid. Also authors should mention about detail method of oil red O staining, how to measure adipose area (use imaging software?), and how to measure body fat percentage shown in result.
10. Figures. Letters is too small to see (group name and letters that show significant difference are also too small).
11. Figure 1D. Both TA and body size may affect food intake. Would you mind to show food intake week by week.
12. Figure 2J. Generally, HFD induced hypertrophy of pancreatic islet. Please show distribution of islet size and compare those.
13. Please inform what does AKP indicate?
14. Figure 3 please indicate inflammatory infiltration structure in picture by arrow.
Reviewer 3 Report
Interesting research on a kind of ignored area which is now quickly gaining increased attention. The study set-up is clear and straightforward and relevant analyses have been performed. The manuscript over-all is clear and well-written, but there I do have some remarks:
- although the English language is correct some formulations could be improved: As an example r45: "the disorder of gut microbiota". I guess the authors want to say a link between obesity and gut microbiota composition has been described, so just make this clearer
- Fig 1B: the signs for differentiating between the diets should be made more distinctive. Even with significant magnification I still have to guess which line represents which diet
- Microbiota results have been analyzed by standard methods and results are shown. Interpretation of these results however can be improved and discussed in more detail. At the phylum level increases is Firmicutes and Proteobacteria are observed, but not at the genus level. What does this mean? What is the role of Desulfovibrio in this? How clear is the assocation between microbiota composition and the tannin effects overall?
- One potential explanation for the observed results is currently lacking: what is not microbiota composition but microbiota activity is responsible for the tannin effect? There is growing evidence that microbiota activity is far more important than its composition, so this should be discussed as well
